# U-net Ensemble Model for Segmentation in Histopathology Images

Yilong Li, Xingru Huang, Yaqi Wang,
Zhaoyang Xu, Yibao Sun, and Qianni Zhang

Graduate School of Computer Science, Queen Mary University of London, London,
UK, yilong.li@qmul.ac.uk

**Abstract.** In this work, a multi-scale U-net[1] fusion model is proposed
for the automatic cancer detection and classification in whole-slide lung
histopathology[2]. The model integrates two types of U-net structure,
trained on different image scales and subsets, aiming to address the challenges posed by the significant variation in data presentation. Since lung
histopathology images come in various sub-categories and appearances,
the performance of an individual trained network is usually limited. We
train a variety of networks by using multiple re-scaled images and different subsets of images, and finally ensemble the outputs of various
networks. Smoothing and noise elimination are conducted using convolutional Conditional Random Fields (CRFs)[3]. The proposed model is
validated on Automatic Cancer Detection and Classification in Wholeslide Lung Histopathology (ACDC@LungHP) challenge in ISBI2019. Our
method achieves a dice coefficient of 0.7968, Which is ranked at the third
place on the board.

**Keywords:** Model ensemble, Tumor, Segmentation, Convolutional CRFs

## 1 Introduction

Digital pathology has been gradually introduced into clinical practice. Digital
pathology scanners can provide whole slide images (WSI) with very high resolution, but for pathologists, time and energy required to manual analysis on WSI of
every single case are unbearable. Automatic computational analysis algorithms
based on machine learning provides a way to reduce pathologists' workload. This
project focuses on the detection and segmentation of lung tumour tissues.[4] Currently, most types of lung cancer are analysed mainly by pathologists' naked eye
examination of the slice image of pulmonary lobes to determine the location,
size and pattern of the tumor. This kind of diagnosis plays a vital role in the
prognosis of lung cancer and the determination of the therapeutic regimen. However, because of the massive number of pathological slices, and too much time
spent on every pathological slice, the process of the diagnostic method is usually tricky. Therefore, segmenting lung tumors through automatic analysis of
pathological slices can help pathologists save significant time and efforts. Deep
convolutional learning method based on neural network is currently the most

popular technology in dealing with the problems of tumor segmentation. However, the network-based training method requires a huge number of datasets to validate and adjust the parameters of each convolutional layer in the convolutional network, to achieve accurate tumor segmentation. Therefore, the primary purpose of this study is to use limited datasets, through selection and optimisation of single networks or combining multiple networks, to achieve efficient and more accurate segmentation of tumor in pathological image slices of lung cancer.

In this work, we design a specific process of tumor tissue segmentation. The dataset is trained to generate the probability map of the mask of the WSI image. Training on the varying data will mislead the model and as a result, produce unsatisfactory prediction results. Hence, we believe that the employment of multiple specifically trained models is necessary. Empirically, a well-trained model can identify the location of the tumor if magnification of at least 10X is used. Thus, the dataset is classified into three sub-datasets by k-mean algorithm.[5] And each sub-dataset is trained with the modified U-net network. In this way, all of the data are used in training, and the different features can be reflected on the different model. This method can analyse the features of the WSI image more specifically so that the difference between different tumor tissue and different sources of the tumor can be studied by the models. Then, with the model ensemble of all the models trained by whole dataset or the three sub-dataset, the performance of the ensemble model is much better than any single model we trained. Finally, image smoothing and noise elimination are conducted after training by using convolutional Conditional Random Fields (CRFs), and the output images are aligned with the original images. The output of convolutional CRFs is the final output of the experiment.

The design of the overall model structure is illustrated in Fig.1.

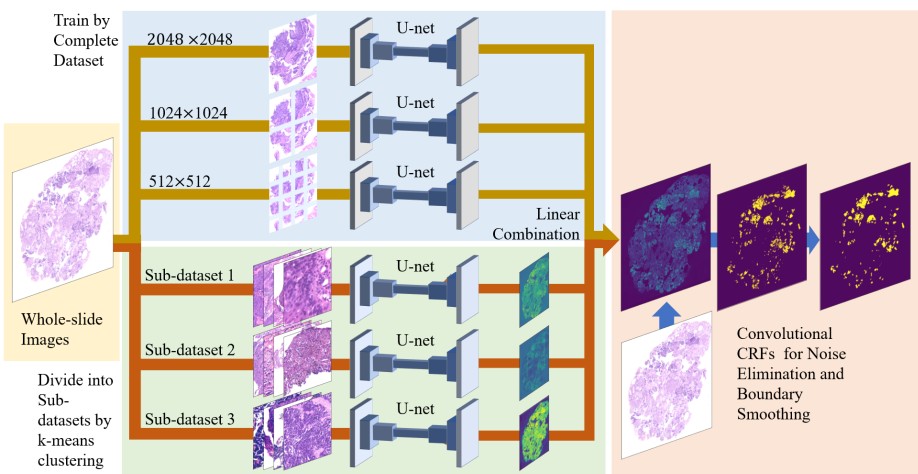

**Fig. 1.** The overall framework of tumor tissue segmentation.

## 2   Methods

In this work, there are three crucial processes of tumor segmentation. The first one is data preprocessing. The dataset is classified into three sub-dataset by the k-means clustering algorithm. Then, with the difference of resolution of the dataset and the difference of sub-datasets, two types of six models in total are designed and trained. Then, fusing multiple models and merging several types of masks, more intuitively accurate segmentation masks are generated.[6] Finally, the convolutional CRFs is used to eliminate the noise and smooth the boundary of the segmentation mask.

In Fig.1, there are six models in total, model 1-3 are trained with the whole dataset, but with three resolutions of 576,1152 and 2048 pixels, model 4-6 are trained with the sub-datasets. The datasets are divided into three sub-datasets by k-means algorithm, so model 4-6 trains with the sub-datasets 1,2 and 3.

### 2.1   Model Training

The overall objective of the proposed network is to make the segmentation of the tumor tissue from normal tissue in the WSI image. Due to the purpose, a modified U-net is used to focus the model more on the overview.

In this network, the energy function is computed by a pixel-wise soft-max[7] over the final feature map combined with the cross entropy loss function.[8]

The soft-max is defined as:

$$p_k(x) = exp(a_k(x))/(\sum_{k'=1}^{K} exp(a_{k'}(x))) \tag{1}$$

where $p_k(x)$ denotes the probability value in feature channel k at the pixel position $x \in \Omega$ with $\Omega \subset \mathbb{Z}^2$. $K$ is the number of classes and $p_k(x)$ is the approximated maximum-function. i.e. $p_k(x) \approx 1$ for the $k$ that has the maximum activation $a_k(x)$ and $p_k(x) \approx 0$ for all other $k$.

The probability value for each epoch should be influenced by the loss function and the result of this influence will reflect in next epoch.

The cross entropy then penalizes at each position the deviation of $\ell(x)$ using

$$E = \sum_{x \in \Omega} w(x)log(p_{\ell(x)}(x)) \tag{2}$$

where $\ell : \Omega \to \{1, ..., k\}$ is the true label of each pixel and $w : \Omega \to \mathbb{R}$ is a weight map that we introduced to give some pixels more importance in the training.

The separation border is computed using morphological operations. The weight map is then computed as

$$w(x) = w_c(x) + w_0 \cdot exp(-\frac{(d_1(x) + d_2(x))^2}{2\sigma^2}) \tag{3}$$

where $w_c : \Omega \to \mathbb{R}$ is the weight map to balance the class frequencies, $d_1 : \Omega \to \mathbb{R}$ denotes the distance to the border of the nearest cell and $d_2 : \Omega \to R$ the distance

to the border of the second nearest cell. In our experiments we set $w_0 = 10$ and $\sigma \approx 5$ pixels.

The out put of the network for each training epoch in this work should be the probability matrix of segmentation mask, so the matrix $\mathbf{P}$ should be

$$\mathbf{P} = [p_k(x)]_{m,n} \tag{4}$$

where $x$ is the coordinate of each pixel in the image,$m$,$n$ is the size of the image.

## 2.2   Model Ensemble

As shown in the Fig.1, there are totally six models of two types. Model 1-3 are trained by the whole dataset, which recalls the performance of the coarse level model. Model 4-6, however, is nearly opposite to the first three models. In that way, one would consider fusing the output of the two levels appropriately, so that they complement each other and jointly explore the advantages.

For the fine level of networks, three sub-datasets are roughly identified based on k-means algorithm. The algorithm calculate the mean of the color value[?] in tumor region for each image based on the true mask.

Because the total variance is constant, this is equivalent to maximizing the sum of squared deviations between points in different clusters (between-cluster sum of squares), which follows from the law of total variance. Based on the three datasets generated by k-means algorithm, three model are generated by the network(model 4-6 in Fig.1),these six models perform well in different situation, so the model ensemble is necessary in this work. The model ensemble function is defined as:

$$\mathbf{F} = \sum_{i=1}^{n} W_i * \mathbf{P}_i \tag{5}$$

Where $\mathbf{F}$ is the output matrix of the model ensemble, $n$ represents the number of models, $W_i$ represents the weight of model $i$, and $\mathbf{P}_i$ (from formula 4) represents the prediction probability map of model $i$.

## 2.3   Convolutional CRFs for Semantic Segmentation

With the model ensemble, the generated masks are clearer, but some noise remains. Because the mask boundaries are mainly generated using 512*512 patches, the predicted model is very sharp. Due to the defect of the above method, the generated boundaries by the coarse level networks lack precise details, and the fine level networks are often attracted by outlier cells, leading to noise.

Also, the isolated tumor cells and fine details in boundaries are often not considered in human manual labeling. Thus, a post-processing step using convolutional CRF is introduced here to further improve the matching of the generated results to the manually annotated masks.

All parameters of the convolutional CRFs can easily be optimised using backpropagation. Semantic image segmentation, which aims to produce a categorical

label for each pixel in an image, is a very import task for visual perception. At the same time, CRFs can also be used to extract the features of the tumor boundary, to optimise the mask generated by the model ensemble, and optimise the mask boundary to make it closer to the hand-drawn mask.

## 3 Experiments

### 3.1 Dataset and Data Preprocessing

**Dataset**.     The dataset comes from the challenge on ACDC@LungHP in ISBI conference 2019. 200 whole-slide images are adopted, 150 of which are used as training sets. It is noted that the sources of the training sets are similar, but the image present significant differences for several reasons. For example, there are obvious differences in the ways of staining, and some pictures even present uneven coloring.

**Data Preprocessing**.     In this work, the dataset is separate into three sub-datasets in order to train the models for model ensemble.

By k-means algorithm, the dataset is classified by the color value of the tumor region, these three sub-datasets are presented in Fig. 2.

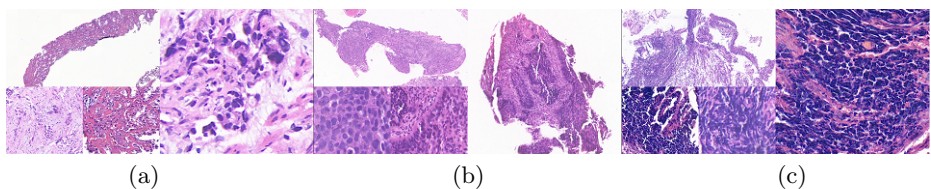

          (a)                   (b)                  (c)

**Fig. 2.** Examples of three identified subsets (**a**) Datasets with lowest color value in tumor region by k-means algorithm, pink tissue around the nucleis in the tumor region, (**b**) Datasets with medium color value in tumor region, light purple tissue around the nucleis in the tumor region, (**c**) Datasets with highest color value in tumor region, deep purple tumor tissue in the tumor region due to the existence of necrosis or some staining mistakes

### 3.2 Training details

**Model Ensemble**.     In order to fulfil the algorithm, the six models should be generated. The model 1-3 are trained by all WSI image in the dataset. It turns out that these three networks show superior ability in noise elimination and pinpointing the location of tumor regions more precisely than the deeper networks in certain images. However, this type of networks can not produce detailed masks on their own, so the fine level of networks are combined for their complimentary expertise on the definition of details.

The model 4-6 are trained with three sub-datasets classified by the k-means algorithm. Thus a deeper U-net is used for detailed classification. At the same time, the level0 images of 512*512 are adopted to guarantee the details and a certain field of vision. It is observed that these three networks have completely different effects on the same picture after training.

The difference for model 1-3 is the resolution of the training image, so the model ensemble for model 1-3 is not necessary. In that way, we first try the model ensemble of 4-6 which properly stands for the whole dataset in total.

The masks generated by six model focus more on the tumor tissue; it has a much lower possibility value on non-tumor tissue region which makes the segmentation more precisely.

Naturally, one would consider fusing the output of the two levels appropriately, so that they complement each other and jointly explore the advantages, i.e. improving the accuracy while at the same time, have noise eliminated.

**Convolutional CRFs**.      At this time, the generated masks are clearer, but some noise remains. Because the mask boundaries are mainly generated using 512*512 patches, the predicted model is very sharp. Due to the defect of the above method, the generated boundaries by the coarse level networks lack precise details, and the fine level networks are often attracted by outlier cells, leading to noise. Also, the isolated tumor cells and fine details in boundaries are often not considered in human manual labeling.

Thus, a post-processing step using convolutional CRF to further improve the matching of the generated results to the manually annotated masks. The resulting mask is bounded by the actual mask edge. The majority of the noise is removed, and smoother edges can be obtained, which are more coherent to those on human annotated masks. The result of model ensemble and the comparison of the mask before and after using convolutional CRFs are shown in Fig.3.

### 3.3   Results

In the challenge of ACDC@LungHP, dice coefficient[**?**] is used as the evaluation metric.

$$DSC = \frac{2\,|X \cap Y|}{|X| + |Y|} \tag{6}$$

The model we propose achieved 0.7968 of dice coefficient, ranked at the third place on the board. Despite the good evaluation, the accuracy is still affected by the wrong classification of subsets and the different nature of manual and generated masks. The data set of this challenge can also be used for classifying the main lung cancer sub-types. This means that the structures and morphology of the different sub-types of tumor tissues vary widely and are complicated. Therefore, each network for a sub-type is likely to be biased towards its sub-category of the tumor and insensitive to other sub-categories. Thus, if the preliminary classification is inaccurate, the performance of the resulting specifically trained

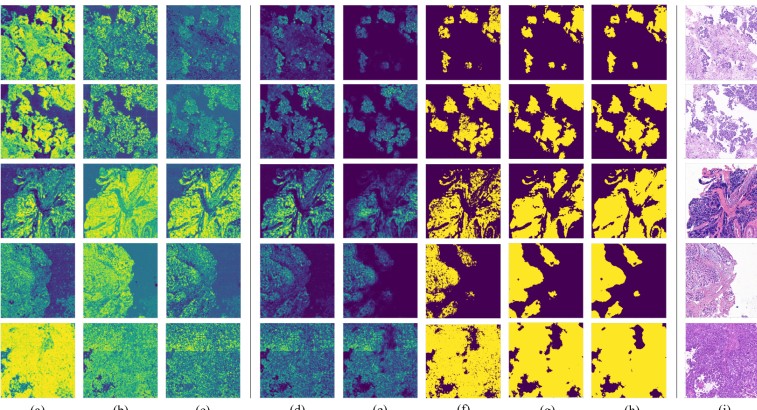

**Fig. 3.** Model ensemble and convolutional CRFs results(image is the test data of ACDC@LungHP) (**a**) Result of the single U-net with 512 pixels' resolution images(model 1), (**b**) Result of the single U-net with 1152 pixels' resolution images(model 2), (**c**) Result of the single U-net with 2048 pixels' resolution images(model 3), (**d**) Result of the combined masks from the three fine level networks,model 4-6 from figure 1, (**e**) Result of the combined masks of fine and coarse level networks,model 1-3 combined with model 4-6 from figure 1, (**f**) The generated mask before using convolutional CRFs, (**g**) The generated mask using convolutional CRFs with a low threshold, (**h**) The generated mask using convolutional CRFs with a high threshold, (**i**) The original WSI provided as reference

model will be affected. Such errors are inevitably propagated to the fused model and reflected in the final output.

Furthermore, the dice coefficient score can be easily affected by the selection of the threshold. Table 1 shows the evaluation results of different runs, using different setups of our model. In the table, the single U-net stands for the model 1-3 in Fig.1, sub-dataset U-net stands for the model 4-6 in Fig.1 which are the network for the sub-datasets. The resulting mask is bounded by the actual mask edge. The majority of the noise is removed, and smoother edges can be obtained, which are more coherent to those on human annotated masks.

| Method | Dice Coefficient (Highest) |
| --- | --- |
| Single U-net (512) | 0.687 |
| Sub-dataset1 U-net | 0.706 |
| Fusion | 0.769 |
| **Fusion + ConvolutionalCRF** | **0.797** |

**Table 1.** Raw results based on multiple methods

In table 1, the best result of single U-net was from model 1 which used the whole dataset with 512 pixels' resolution. The best result of single sub-dataset U-net was from model 4 which used the first part of the dataset. The best model ensemble result was the result of the model ensemble of all six models. The best result of all the method was the model ensemble of all six models and the method of convolutional CRFs, and this result was ranked at the third place on the board of the ACDC@LungHP challenge in ISBI2019.

## 4    Conclusions

We propose an automatic cancer detection and classification method based on U-net and an ensemble scheme of multiple networks leading to a merged mask for segmentation. It is shown that using such a fusion model, more accurate segmentation can be acquired compared to relying on a single network. The result shows that when dealing with complex data sets, multiple networks fusion demonstrates an evident advantage. Convolutional CRFs for noise reduction and tumor border smoothing further enhance the boundary accuracy. Moreover, as reflected by the success of the combined multi-network model, our specifically trained networks for different sub tissue types provide a good foundation for stage two challenges on the lung cancer subtype classification.

For future work, the performance of the multi-network fusion model will be further improved by adding a self-learning classifier, if cell type labels can be introduced. At the same time, a better classification model will be developed for preliminary sub-type classification.

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
