# OpenReview forum: "U-net Ensemble Model for Segmentation inHistopathology Images"
_MICCAI.org/2019/Workshop/COMPAY — Submitted to COMPAY 2019_

### Official Review · AnonReviewer2 · 2019-08-05
**Lung cancer in whole-slide histopathology images is detected using six different models derived by training the U-Net architecture on six different data representations of scale and data variability. Overall, the paper is nicely written (though some minor language errors could be corrected for). However, there are some major concerns regarding the technical choices in the proposed method and evaluation of results which are commented below.**

**Rating:** 5
**Confidence:** 5

**Review:**

Major concerns
Different combinations of models are ensembled to evaluate and compare the results of the proposed approach. Conclusively, this work demonstrates a nice application of CNNs in computational analysis of large WSI images to assist pathologists and thereby indicating the potential usefulness of the CNNs’ for clinical purposes. The fused output is further refined and smoothed using convolutional Conditional Random Fields and validated on dataset provided in ACDC@LungHP challenge at ISBI19, and reported results are ranked at the third place.

1.	A major concern is regarding the evaluation of the proposed approach. Since authors have used the complete dataset i.e., 200 WSI images for training and evaluation of the models. This could potentially introduce bias in the evaluation. Please explain why the evaluation is not reported on the test data provided by the challenge organizers?
2.	It is unclear how the U-Net is modified across the models, apart from being trained of data with different resolution or color. Any other modifications in relation to the original U-Net should also be described.

3.	The authors report only the results of a single sub-model with 512x512 dimensions instead of all other sub-models. Same implies to the other remaining models trained with sub-datasets. To demonstrate the overall impact of the proposed approach, it is crucial to report the results of each individual sub-model for a fair comparison.

4.	It is stated the mean color value used for k-means clustering is calculated ‘in the tumor region for each image based on the true mask’: How will this be implemented when using the model to classify real data where no mask is available. Please also motivate the choice of three clusters.

5.	Another concern is regarding the values of weight w0 and sigma (page 4). How are these values computed? Also, how can the same value of sigma compensate for different scales?

6.	In figure 1, please clarify what the actual pixel dimensions are of the data that is input to the network. For example, is model 1 trained with patches of the same size as model 3, but sampled differently? The authors state that the difference for model 1-3 is the resolution of the training image, so the model ensemble for model 1-3 is not necessary.  If so, why do they applying it? Since the U-Net architecture can already cater scale variance, what is the motivation of using three different streams based on scales? Please justify why you opted this particular strategy to design the methodology.





Minor edits

7.	In the introduction, “pathologists’naked eye” is likely not true as a microscope would be used. Please revise.
8.	Page 2: There is a typo in the 2nd line of the 2nd paragraph where “dataset” should be replaced with ‘network models’.
9.	Page 2: What do authors imply with the statement “different sources of the tumour can be studied by the models”. It will be nice to explain how the fused or even single model can compensate for the different sources of tumour when the sources of the training sets are already similar as of stated in the Section 3.1.
10.	Page 2: Please explain how the the output images are aligned with the original images. Also, how the outputs of the individual models 1-3 are linearly combined when there is scale variance among the outputs.
11.	Page 3: Section 2: What criteria is considered to categorize the dataset into three subsets? Also, do authors imply difference of scales when referring to as difference of resolution? Please consider revising the statement for clarity.
12.	Please describe briefly what the ground truth is and how it was created.
13.	Page 3 Section 2: There are typos in the size dimensions of three scales. Please check it throughout the manuscript.
14.	Page 3 Section 2.1: Please describe which distances (d1 and d2) are authors referring to as in equation 3. Also, are these distances calculated from the border or centre of the cells to the border of the nearest cells and border of the nearest cells, respectively.
15.	The reference for color value calculation is missing.
16.	Page 4: Section 2.2: Please state how the training of the ensemble model is performed. Explain whether the network models were trained separately for the fusion or if the training was performed in end-to-end fashion? Also, what criteria are considered to decide the weight threshold for each individual network when performing the fusion?
17.	Page 4: Section 2.2: Please clarify which different situations you are referring to as when these six modes performed well. Also, please check and fill in the missing colour value.
18.	Page 4: Section 2.3: Do authors imply the predicted output of the model instead of “predicted model”? Please revise it accordingly.
19.	Page 5: Section 3.1: Have the authors considered to perform stain normalization to compensate for the data heterogeneity Please discuss.
20.	Page 6: What does level0 images imply here?
21.	Page 6: Section 3.3: Reference to the dice coefficient is missing.

---

### Official Review · AnonReviewer4 · 2019-08-15

**Rating:** 3
**Confidence:** 3

**Review:**

This paper describes the entry that had the third best score in the ACDC challenge on lung histopathology classification. The methodology is based on an ensemble of different U-Net models. While the overall methodology seems reasonable, I have some concerns regarding the writing style of the paper, motivation for the method design and evaluation of the different components.

While the paper can relatively easily be understood, the writing style is rather poor for a scientific publication. For example, “unbearable” and “tricky” is a rather dramatic way to describe the workload and work of pathologists. The authors spend much space on textbook-like description of know concepts such as softmax, while other potentially more important aspects are not well explained. For example, it is not clear how the U-Net was modified. At a few places there are missing references (“[?]”).

I am a bit puzzled why a CRF postprocessing step is needed since the U-Net architecture already accounts for spatial context information. This has me suspecting that the training of the U-Net models was not done in an optimal way. In the results table, the authors should specify the performance of all individual U-Net before the fusion.

“Because the mask boundaries are mainly generated using 512*512 patches, the predicted model is very sharp.” It is not clear why this is the case. Does this mean that the other networks in the ensemble have limited utility?